# Straw Biochar-Facilitated Methanogenesis from Acetic Acid and Ethanol: Correlation with Electron Exchange Capacity

Yannan Ruan [1], Yuze Jiang [1], Moting Li [2], Suyun Xu [1,*], Jining Zhang [3], Xuefeng Zhu [1] and Hongbo Liu [1,*]

[1] School of Environment and Architecture, University of Shanghai for Science and Technology, Shanghai 200093, China; yannanruan1997@126.com (Y.R.); yuzejiang2023@126.com (Y.J.); xuefengzhu@usst.edu.cn (X.Z.)

[2] Shanghai Institute of Mechanical & Electrical Engineering Co., Ltd., Shanghai 200040, China; limt@shanghai-electric.com

[3] Eco-Environmental Protection Research Institute, Shanghai Academy of Agricultural Sciences, Shanghai 201106, China; j.n.zhang@163.com

* Correspondence: xusy@usst.edu.cn (S.X.); liuhb@usst.edu.cn (H.L.)

**Abstract:** Straw biochar prepared by three methods (i.e., pyrochar, $HNO_3$-modified pyrochar, and hydrochar) was added to the anaerobic digestion system with acetic acid and ethanol as substrates to explore the effects of biochar on methane production, substrate degradation, and microbial community structure. The biogas yields of the biochar-supplemented groups all increased, and the maximum methane yield was found in the hydrochar group, which was 45.4% higher than the control. In the ethanol-fed reactor, the maximum partial pressure of hydrogen in the headspace of the hydrochar reactor was reduced from 3.5% (blank reactor) to 1.9%. Overall, methane production is directly proportional to the electron exchange capacity (EEC) value of biochar. Furthermore, the bio-aging process increased the EEC of each kind of biochar to 5.5–8.1%, which was favorable for the sustainable promotion of methanogenesis. The increased methane yield from the bio-aged biochar could either be attributable to the changes in surface oxygen-containing functional groups or the selectively enriched microbial community on the biochar, such as *Geobacter*, which could participate in direct interspecies electron transfer.

**Keywords:** anaerobic digestion; biochar; electron exchange capacities; ethanol; hydrochar

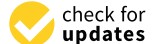



## 1. Introduction

Worldwide, the extensive use of fossil fuels will lead to a series of environmental problems, such as the greenhouse effect [1], air pollution [2], acid rain [3], and so on. Statistics from the International Agricultural and Food-related Organizations show that one-fifth of the world's agricultural straw is produced in China [4]. In recent years, although the utilization rate of agricultural straws has increased China due to governmental guidelines and related industry development, a large amount of straw is either directly returned to the field or utilized as fertilizer. With the gradual improvements in the requirements of carbon neutral management in agriculture, the need to develop high-value utilization has become increasingly prominent [5]. The recycling strategies of cellulosic biomass can be divided into two categories: one is as an alternative fuel with huge energy application potential, and the other is to develop carbon-based materials, such as the generation of biochar via the pyrolysis process [6].

Biochar prepared from agricultural biomass is characterized by a strong adsorption capacity, rich chemical composition, economic efficiency, and environmental friendliness, making it an important cell colonization carrier [7]. In recent years, many papers have explored the potentiality of biochar as an anaerobic digestion improver for biological fertility, facilitating direct interspecific electron transfer (DIET) within methanogenic symbiosis [7,8]. Anaerobic digestion refers to the digestion technology in which facultative bacteria and

anaerobic bacteria decompose biodegradable organic matter into $CH_4$ and $CO_2$ [9]. Anaerobic digestion is widely used in organic solid waste treatment, which can achieve the development of a circular economy, environmental protection, reducing greenhouse gas emissions, producing renewable energy, and other goals. However, an anaerobic digestion system also has many disadvantages, such as its slow reaction rate and low methane yield.

Recently, a growing number of researchers have become concerned about the correlations between material properties and biochar-mediated enhancements of methane production [10]. It is noted that characteristics such as conductivity, porosity, and redox property were successively proposed to be important factors in enhancing methanogenesis [11]. Comparatively, due to the existence of redox-active organic functional groups in biochar, the electron exchange capacity (ECC) of biochar proved to be vital for the enrichment of electroactive groups and the establishment of syntrophic relationships between different microbial species [12]. Meanwhile, expanding the surface area can effectively improve the electron transfer capacity of biochar and expose more redox-active groups, so as to obtain a better redox activation performance [10]. Zhang et al. [13] found that the graphitic structure in biochar, formed under high-temperature pyrolysis, could enhance the biological redox reaction by shuttling electrons. When biochar was modified with the surface oxidation of $HNO_3$ or $H_2O_2$, the phenolic and lactonic groups on biochar were increased, leading to a higher value of electron-donating capacity (EDC) [11]. Magnesium-modified coconut shell biochar was also found to have an improved ion exchange capacity, mineral precipitation, and oxygen functional group interactions [14]. Chen et al. [15] found that methanogenic co-cultures in the presence of biochar were able to utilize 86% of the electrons released from ethanol oxidation toward methane production, but in the absence of biochar, no ethanol was consumed and no methane was produced. Yamada et al. [16] found that the supplementation of magnetite induced electric syntropy between organic acid-oxidizing bacteria and methanogenic archaea and accelerated methanogenesis, even under thermophilic conditions.

Many of the present studies on this topic operate in a single-batch mode to reveal the short-term effects of biochar [17]. In fact, during the long-term operation, both the microbial symbiosis and physiochemical properties of biochars will change accordingly [18,19]. Biochar aging refers to the process of changing the properties of biochar under the influence of various abiotic and biotic factors, such as precipitation events, temperature, oxidation, and biodegradation [20]. The contents of O-containing functional groups like C–O, –OH, and C=O increased with the freeze–thaw cycling treatment and alternating dry-wet aging treatment; nevertheless, natural aging with fresh soil at 25 °C for 30 days only exerted a few changes on physicochemical properties of wheat straw biochar [21].

In this study, straw biochars obtained using different preparation methods (i.e., hydrothermal carbonization, pyrolytic carbonization, and pyrolytic carbonization+ $HNO_3$ modification) and the bio-aging process were characterized with EEC values and their differentiated effects on anaerobic digestion with acetate and/or ethanol. The characteristics of gas production, substrate degradation, and microbial community changes were detected to explore the influence of biochar on the methane production process of anaerobic digestion and analyze the effect of biochar EEC on promoting anaerobic digestion efficiency.

## 2. Materials and Methods

### 2.1. Preparation of Straw Biochar

Three kinds of biochar were prepared, i.e., 235 °C hydrothermal carbonization (Hy), 400 °C pyrolytic carbonization (Py), and 400 °C pyrolytic carbonization + $HNO_3$ modification (Mo), respectively. These three kinds of biochar are classified as the "new biochar group (N)". Another group of biochar named "bio-aged biochar group (B)" was collected from the anaerobic digesters supplemented with each kind of biochar after three batches of operation. The co-culture of anaerobic sludge and biochar enabled the biological aging process for biochar at 35 °C.

### 2.2. Experimental Design

During the anaerobic digestion experiment, the determined amount of inoculum and substrate was put into a 600 mL anaerobic digestion flask with a working volume of 400 mL. The concentration of biochar was 10 g/L, and the total solid content of anaerobic sludge was 0.72%. A group of blank control experiment (CK) without biochar additions was set up. Acetic acid and ethanol were sequentially used in this experiment, which was divided into four experiments. The initial concentrations of acetic acid and ethanol were sequentially set up as 0.6 g/L ethanol and 0.6 g/L acetic acid (AE0.6), 1.2 g/L ethanol and 1.2 g/L acetic acid (AE1.2), 1.2 g/L ethanol (E1.2), and 2.4 g/L ethanol (E2.4), respectively, for a four-batch study. Both suspended sludge and attached biofilm samples were collected from the reactors after the anaerobic digestion test to characterize the microbial community (Table 1).

**Table 1.** Naming rules for sludge samples.

| Biochar Types | Sampling Method | Name |
|---|---|---|
| Pyrochar (New) | Attached biofilm | Py-N |
| Modified (New) | Attached biofilm | Mo-N |
| Hydrochar (New) | Attached biofilm | Hy-N |
| Pyrochar (New) | Suspended sludge | Py-NS |
| Modified (New) | Suspended sludge | Mo-NS |
| Hydrochar (New) | Suspended sludge | Hy-NS |
| Pyrochar (Bio-aged) | Attached biofilm | Py-B |
| Modified (Bio-aged) | Attached biofilm | Mo-B |
| Hydrochar (Bio-aged) | Attached biofilm | Hy-B |
| Pyrochar (Bio-aged) | Suspended sludge | Py-BS |
| Modified (Bio-aged) | Suspended sludge | Mo-BS |
| Hydrochar (Bio-aged) | Suspended sludge | Hy-BS |

### 2.3. Analytical Methods

The concentrations of acetic acid and ethanol were determined by Waters 2695/2489 high-performance liquid chromatography [22]. The gas generated from each reactor was collected in a gas bag, and its volume was measured with a syringe. The methane content of biogas was measured by gas chromatography (GC2014C, Shimadzu, Japan) with a thermal conductivity detector, and helium gas was used as the carrier gas. The EEC value of biochar was determined using mediated electrochemical oxidation (MEO) and electrochemical reduction (MER) methods, respectively, to determine the EDC and EAC of biochar [23].

Kinetic analysis of anaerobic digestion methane production: The cumulative methane production in this study can be used for kinetic analysis of anaerobic digestion process with the modified Gompertz Equation (1).

$$\mathrm{B} = B_0 exp\left\{-exp\left[\frac{R_m e}{B_0}((\lambda - t) + 1)\right]\right\} \tag{1}$$

where B is the cumulative methane production (mL-CH$_4$/g substrate). $B_0$ is methane potential (mL-CH$_4$/g substrate). $R_m$ is the maximum methane-producing rate (mLCH$_4$/g substrate D). $\lambda$ is the retardation stage of anaerobic digestion (D). $T$ is the duration of the anaerobic digestion reaction (d). $e$ is a constant (2.718), and parameters $B_0$, $R_m$, and $\lambda$ are used to evaluate the gas production characteristics of biochar-mediated anaerobic digestion. The one-way analysis and paired $t$-test of variance were used to test the significance of the results, and $p$ less than 0.05 was considered a statistical criterion.

The suspended sludge and biofilm attached to biochar were sampled from the digestion reactor after the anaerobic digestion experiment. Illumina Miseq PE300 platform was used for high-throughput sequencing to amplify the V4 region of 16S rRNA. The primers for PCR amplification were 515F (5'-GTG CCAGCM GCC GCG GTA A-3') and 806R (5'-GGA CTA CHV GGG TWT CTA AT-3'). The data from high-throughput sequences were analyzed from the clustering of operational taxonomic units (OTUs) and taxonomy, and

then the OTUs and taxonomy were combined for consistency analysis so as to obtain the basic analysis results of each sample's OTUs and the corresponding taxonomic pedigreage. After basic analysis, statistical and visual analysis were carried out.

## 3. Results

### 3.1. Methane Generation with Different Biochars

3.1.1. Methane Production from Acetate and Ethanol Mixture

The cumulative methane production of biochar-mediated anaerobic digesters is shown in Figure 1. In total, six kinds of biochar were tested, which were classified into two groups, i.e., new biochar (Py-N, Mo-N, and Hy-N) and bio-aged biochar (Py-B, Mo-B, and Hy-B). Sequentially, the mixtures of 0.6 g/L acetate and 0.6 g/L ethanol and 1.2 g/L acetate and 1.2 g/L ethanol were used as the substrates, and the methane yields of each group were recorded and compared.

$$\text{Acetogenesis: } CH_3CH_2OH + H_2O \rightarrow CH_3COOH + 2H_2 \quad (2)$$

$$\text{Methane yield from ethanol: } CH_3CH_2OH + H_2O \rightarrow CH_3COOH + 2H_2 \quad (3)$$

$$\text{Methane yield from acetate: } CH_3COOH \rightarrow CH_4 + CO_2 \quad (4)$$

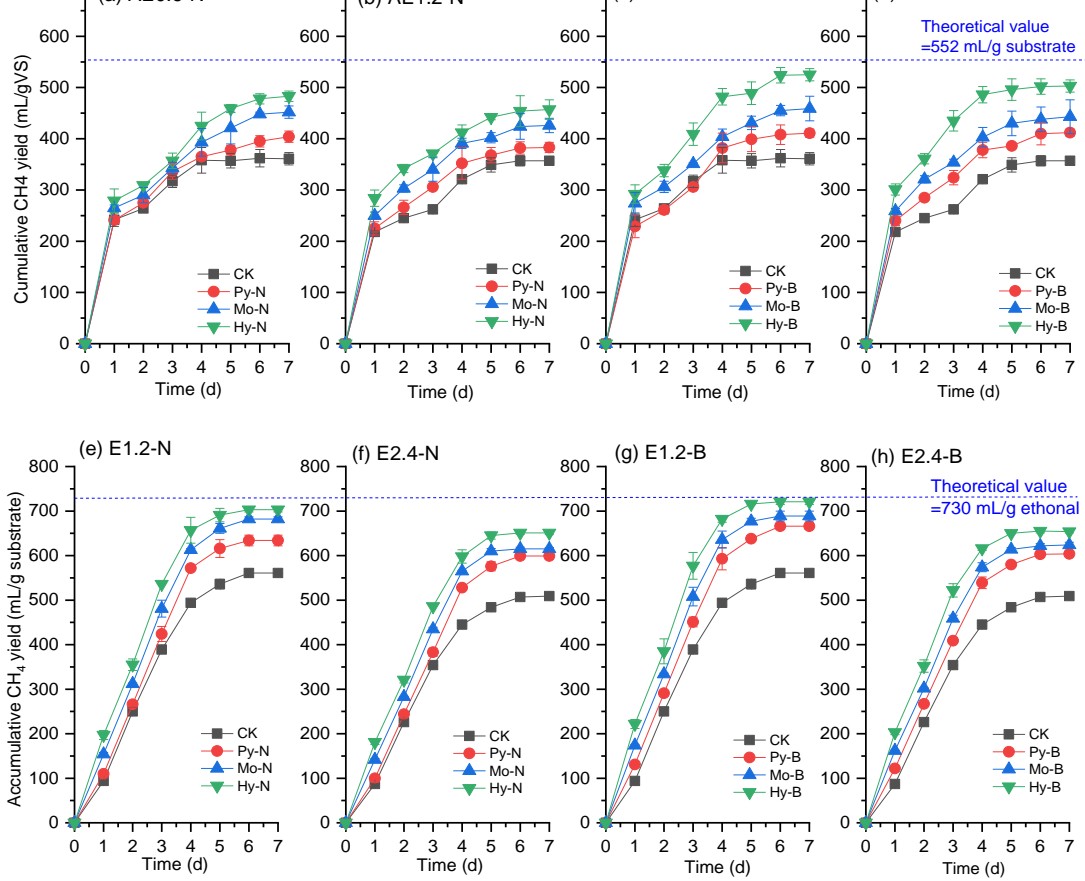

**Figure 1.** Cumulative methane production in batch fermentation mediated by new biochar and bio-aged biochar: (**a**)AE0.6-N, (**b**) AE1.2-N, (**c**)AE0.6-B, (**d**) AE1.2-B, (**e**) E1.2-N, (**f**) E2.4-N, (**g**) E1.2-B, and (**h**) E2.4-B.

The theoretical $CH_4$ yields of acetate and ethanol were 373 mL/g and 730 mL/g, respectively. As the weight ratio of acetate and ethanol was 1:1, the overall theoretical $CH_4$ yields of the acetate and ethanol mixture were 552 mL/g substrate.

For the AE0.6 batch study (Figure 1a,c), the cumulative $CH_4$ yield of the CK group was 361 ± 12 mL/g substrate after 6 days of digestion, and the cumulative $CH_4$ yield of other experimental groups with biochar was 404–525 mL/g substrate. The cumulative $CH_4$ yields were increased by 11.91% to 45.43%. For the AE1.2 batch experiment (Figure 1b,d), the cumulative methane production of the CK group was 357 ± 11 mL/g substrate, and the cumulative $CH_4$ yield of other biochar groups was 383–503 mL/g substrate, which was increased by 7.28–40.9%. Similarly, the maximum $CH_4$ yield was found in the Hy-N and Hy-B groups. The results indicated that the biodegradation and methane production were promoted by biochar addition, and the performance of Hy was superior to other biochar types. This result corresponds to previous reports. The more abundant surface functional groups of hydrochar are favorable for facilitating electron shuttles among syntrophic bacteria [24]. By calculation, the bioconversion rates of the substrate into $CH_4$ in the AE0.6 batch study were 73.2% (Py-N), 74.5% (Py-B), 81.9% (Mo-N), 83.2% (Mo-B), 87.5% (Hy-N), and 95.1% (Hy-B), respectively. Nevertheless, the bioconversion rates in the AE1.2 group all declined, which were 69.40%, 74.65%, 77.19%, 80.27%, 82.81%, and 91.14%, respectively. This result indicated the inhibition of conversion potential under the higher organic loading rate condition.

In addition, the cumulative $CH_4$ yields in the bio-aged biochar group were generally higher than those of the corresponding new biochar group. For example, in AE1.2, the maximum $CH_4$ yields of Hy-N and Hy-B were 483 mL/g substrate and 525 mL/g substrate, respectively. The increased $CH_4$ yield could be attributed to either the stronger bioaffinity toward bacteria or the enhanced syntrophic cooperation. This encouraging result also supported the fact that biochar has the potential for long-term performance and reuse, which can help reduce the cost of biochar supplementation.

### 3.1.2. Methane Production from Ethanol

When ethanol was used as the substrate, the $CH_4$ yield significantly improved compared to the batch study on the acetate and ethanol mixture, which had a theoretical value of 730 mL/g substrate. As shown in Figure 1e,f, the $CH_4$ yield of the control group was 516 ± 6 mL/g substrate after 6 days of digestion. Meanwhile, the cumulative $CH_4$ yields of the experimental groups with biochar supplementation were 634–712 mL/g substrate, which was promoted by 13.01–28.52%. Similar to the AE batch study, the maximum $CH_4$ yield was found in the Hy group. At the same time, the $CH_4$ yield of the Hy-B group (729.1 mL/g substrate) was slightly higher than that of the Hy-N group (745.4 mL/g substrate). The results show that, although the substrate is different, the main influencing factors on methane promotion are basically related to the type of biochar.

### 3.1.3. Kinetics Studies for Methane Yield and Lag Time

The kinetic parameters of biomethanation fitted by the modified Gompertz equation are shown in Table 2. According to the potential methane production ($B_0$), maximum rate of methane production ($R_m$), and time ($\lambda$), the methane production process in four batches of 28 experimental groups was analyzed. For the AE0.6 batch experiment, the maximum methanogenic potential of 568.30 mL/g substrate was obtained in the Hy-B group, which was 50.73% higher than that in the control group. The lag period was shortened in all the experimental groups that were supplemented with biochar. The shortest lag period was found in AE0.6 with Hy-B supplemented, i.e., 0.26 d, which was 45.83% shorter than that of the control group. The lag period of AE1.2 and E1.2 with Hy-B supplementation was 0.32 d, indicating good adapation to the elevated organic load. The E1.2 batch experiment also obtained the maximum methane-producing potential of 745.37 mL/g substrate in the Hy-B experimental group.

**Table 2.** Methanogenic kinetics fitted by modified Gompertz equation.

| The Experimental Group | | $B_0$ (mL/g vs.) | $R_m$ (mL/g vs. d) | $\lambda$ (d) | $R^2$ |
|---|---|---|---|---|---|
| AE0.6 | Blank | 377.03 | 71.31 | 0.48 | 0.9203 |
| | Py-N | 424.84 | 65.66 | 0.44 | 0.9811 |
| | Py-B | 452.66 | 65.13 | 0.42 | 0.9416 |
| | Mo-N | 529.27 | 54.07 | 0.38 | 0.9760 |
| | Mo-B | 527.67 | 56.16 | 0.34 | 0.9797 |
| | Hy-N | 561.38 | 60.85 | 0.29 | 0.9641 |
| | Hy-B | 568.30 | 84.20 | 0.26 | 0.9696 |
| AE1.2 | Blank | 416.94 | 43.45 | 0.53 | 0.9308 |
| | Py-N | 409.92 | 62.74 | 0.45 | 0.9837 |
| | Py-B | 442.90 | 64.07 | 0.41 | 0.9811 |
| | Mo-N | 452.40 | 69.46 | 0.40 | 0.9876 |
| | Mo-B | 467.96 | 75.28 | 0.39 | 0.9876 |
| | Hy-N | 487.45 | 69.58 | 0.38 | 0.9887 |
| | Hy-B | 520.04 | 111.45 | 0.32 | 0.9744 |
| E1.2 | Blank | 577.22 | 168.30 | 0.77 | 0.9987 |
| | Py-N | 660.53 | 188.78 | 0.83 | 0.9921 |
| | Py-B | 693.71 | 188.04 | 0.76 | 0.9943 |
| | Mo-N | 710.38 | 191.94 | 0.66 | 0.9936 |
| | Mo-B | 716.06 | 199.72 | 0.54 | 0.9907 |
| | Hy-N | 729.08 | 204.26 | 0.44 | 0.9886 |
| | Hy-B | 745.37 | 216.32 | 0.32 | 0.9868 |
| E2.4 | Blank | 522.69 | 150.96 | 0.76 | 0.9992 |
| | Py-N | 627.55 | 170.18 | 0.90 | 0.9923 |
| | Py-B | 629.41 | 169.48 | 0.74 | 0.9937 |
| | Mo-N | 643.59 | 177.49 | 0.65 | 0.9878 |
| | Mo-B | 649.14 | 177.75 | 0.53 | 0.9897 |
| | Hy-N | 679.40 | 182.44 | 0.47 | 0.9884 |
| | Hy-B | 676.58 | 194.32 | 0.31 | 0.9882 |

Comparatively, the influence of biochar types on the methane yield was more significant in the batch study with the acetate and ethanol mixture as the substrate. As shown in Figure 2a, the increment of methane yield in the AE-B group was significantly higher than that in the AE-N group ($p = 0.023$); meanwhile, a smaller difference in the methane yield was found between the E-N group and the E-B group ($p = 0.120$). Similarly, the difference among Py, Mo, and Hy was more significant in the AE batch compared to the E batch (Figure 2b).

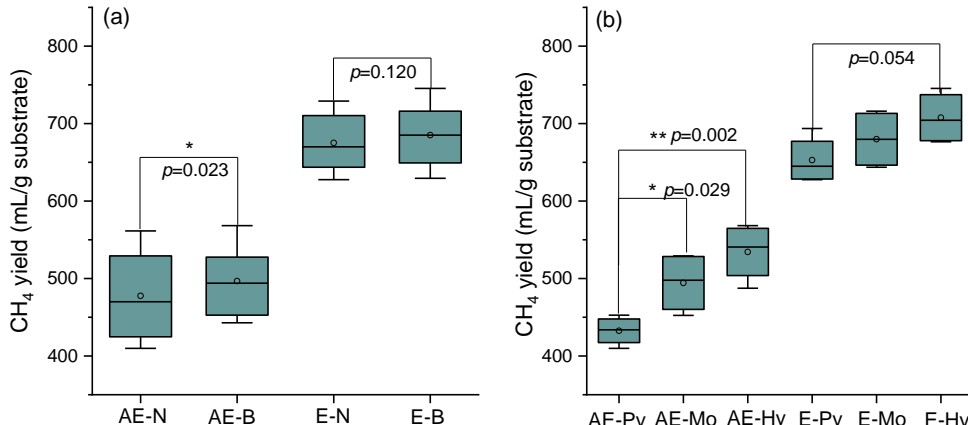

**Figure 2.** Statistical analysis of difference significance on methane yield: (**a**) comparison between different substrate (AE and E) and different type of biochar (new-N and bioaged-B ), (**b**) comparison between different types of biochar (Py, Mo and Hy). * represents $p < 0.05$, ** represents $p < 0.01$.

### 3.2. Substrate Utilization Rate with Different Biochars

#### 3.2.1. Degradation of Acetic Acid and Ethanol

The effect of the biochar addition on the degradation of acetic acid and ethanol during anaerobic digestion is shown in Figure 3. As ethanol is initially utilized by acetogens to generate acetate and $H_2$ for methane production, the accumulation of acetate was observed in the reactors with ethanol. In AE0.6 and AE1.2, 56.5% and 82.0% ethanol from the Hy-B group were consumed, respectively, in the initial two days of anaerobic digestion, while the consumption rates of the control group were only 32.9% and 61.6%, respectively. Obviously, biochar plays an important role in promoting the acetogenesis of ethanol in the anaerobic digestion system. Generally, the difference in the ethanol utilization rate corresponded to the methane yield. These results suggest that straw biochar can promote the co-trophic reaction between acetic acid-producing bacteria and methane-producing archaea, which is conducive to substrate degradation and utilization to enhance methane production.

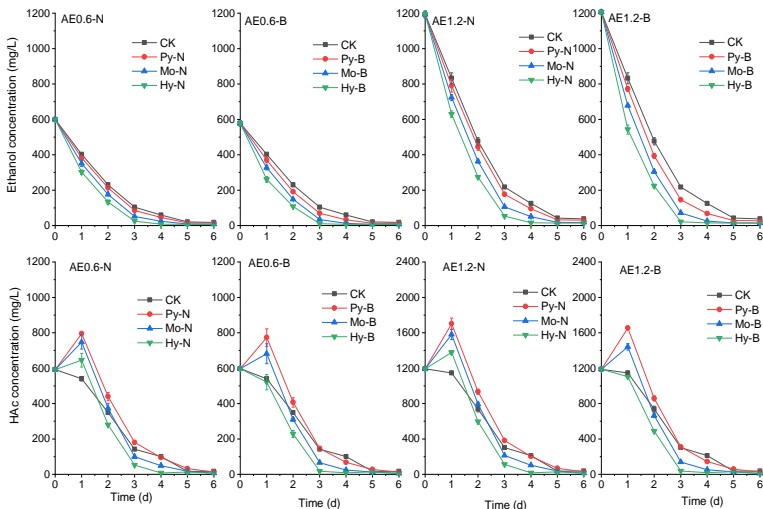

**Figure 3.** Changes in acetic acid and ethanol in AE batch fermentation.

In the batch study of E1.2 and E2.4 (Figure 4), it was found that acetate accumulated during the utilization of single substrate ethanol, although acetic acid is considered to be the most readily available substrate for methanogens. On the one hand, the accumulation of acetic acid is due to its higher production rate than the consumption rate; on the other hand, this may be due to the inhibition of intermediate products, such as the higher partial pressure of $H_2$.

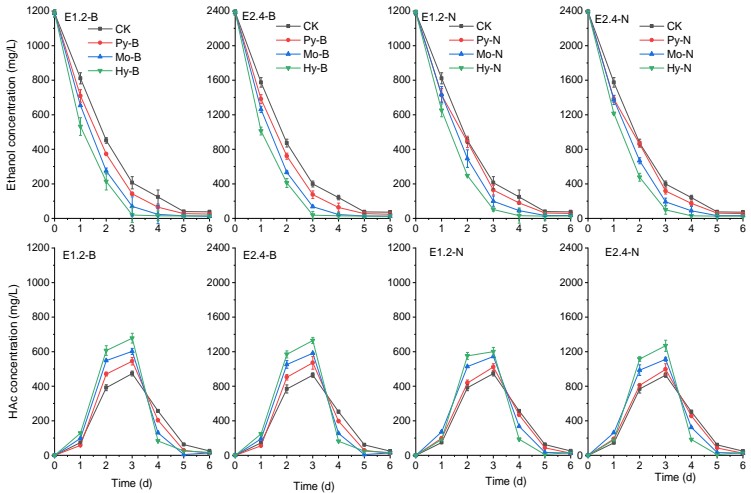

**Figure 4.** Changes in acetic acid and ethanol in E batch fermentation.

### 3.2.2. Change in Hydrogen Content

The complete conversion of ethanol into methane in anaerobic digestion requires the joint action of bacteria and methanoarchaea. Methane production from hydrogen, carbon dioxide consumption, and hydrogen are shown in the following formula:

$$\text{Hydrogenotrophic methanogenesis}: \quad CO_2 + 4H_2 \rightarrow CH_4 + 2H_2O \tag{5}$$

Ethanol supplies electrons:

$$IET: \quad CH_3CH_2OH + H_2O \rightarrow CH_3COO^- + H^+ + 2H_2 \qquad \Delta G_0' = 9.68 \text{kJ/mol} \tag{6}$$

$$DIET: \quad CH_3CH_2OH + H_2O \rightarrow CH_3COO^- + 5H^+ + 4e^- \qquad \Delta G_0' = -149.64 \text{kJ/mol} \tag{7}$$

Converting $CO_2$ to methane:

$$IET: \quad 2H_2 + \frac{1}{2}CO_2 \rightarrow \frac{1}{2}CH_4 + H_2O \qquad \Delta G_0' = -65.35 \text{kJ/mol} \tag{8}$$

$$DIET: \quad 4H^+ + 4e^- + \frac{1}{2}CO_2 \rightarrow \frac{1}{2}CH_4 + H_2O \qquad \Delta G_0' = 93.98 \text{kJ/mol} \tag{9}$$

Acetoclastic methanogenesis:

$$CH_3COO^- + H^+ \rightarrow CH_4 + CO_2 \qquad \Delta G_0' = -35.91 \text{kJ/mol} \tag{10}$$

In E1.2 and E2.4, methanogens degraded ethanol into hydrogen because the electron-supplying bacteria converted ethanol into acetate and released electrons. At the beginning of batch digestion, $H_2$ partial pressure in the headspace increased, but at the same time, it was accompanied by the reaction of hydrogenotrophic methanogenesis to produce methane. The $H_2$ partial pressure of the Hy-B group was maintained at the lowest level (Figure 5). As the batch study of E2.4 was carried out after E1.2, the $H_2$ partial pressure in E2.4 was lower than in E1.2, indicating that the metabolic pathway had undergone adaptive changes. The ratio of hydrogen per day in the experimental group was inversely proportional to its methanogenic performance, indicating that the hydrogen-mediated electron transfer in the experimental group that was mediated by biochar was partially replaced by DIET. This indirectly proved that DIET occurred in the anaerobic digestion process of the biochar-mediated experimental group, and in the hydrothermal carbon-mediated experimental group, DIET had a relatively large contribution rate to ethanol degradation.

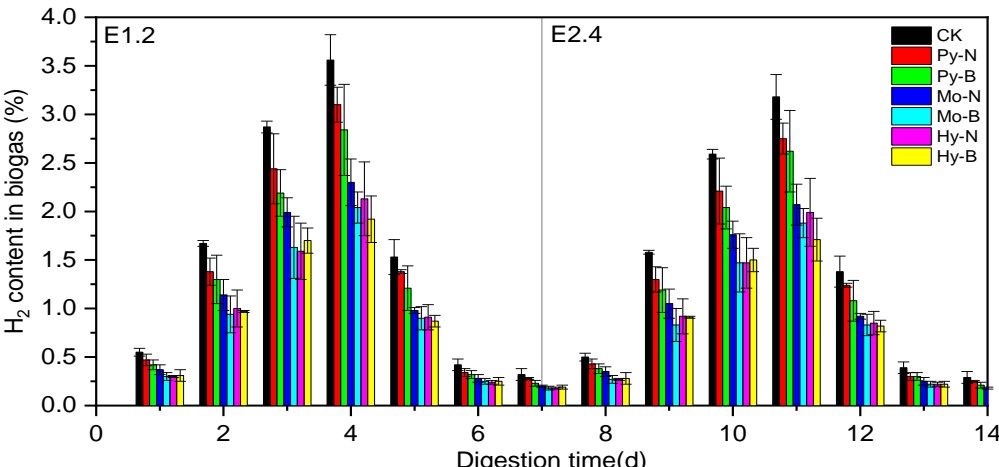

**Figure 5.** Change in hydrogen content in intermediates of biochar-mediated ethanol–methane conversion.

### 3.3. The Correlation Analysis of Biochar EDC and Methane Production Indexes

Through the correlation analysis of EDC and anaerobic digestion indexes in experimental groups with acetic acid and ethanol, it was found that the EDC of biochar had a linear correlation with anaerobic digestion indexes, i.e., the maximum methane yield and lag time (Figure 6). Obviously, the larger the EDC of biochar added into the anaerobic digestion system, the higher the specific methane yield and the shorter the lag period obtained in the experimental group. As the specific methane yields of acetic acid and ethanol are different, the linear fittings were grouped by the substrate type and substrate concentration. The coefficient of determination ($R^2$) between the EDC and the maximum methane yield was 0.914, 0.904, 0.935, and 0.871 for E1.2, E2.4, AE0.6, and AE1.2, respectively. Meanwhile, the $R^2$ value between the EDC and the lag period was 0.958, 0.780, 0.957, and 0.941, respectively. This result indicated that the selection of biochar with a higher EDC value was favorable to promoting methane production in an anaerobic digestion system.

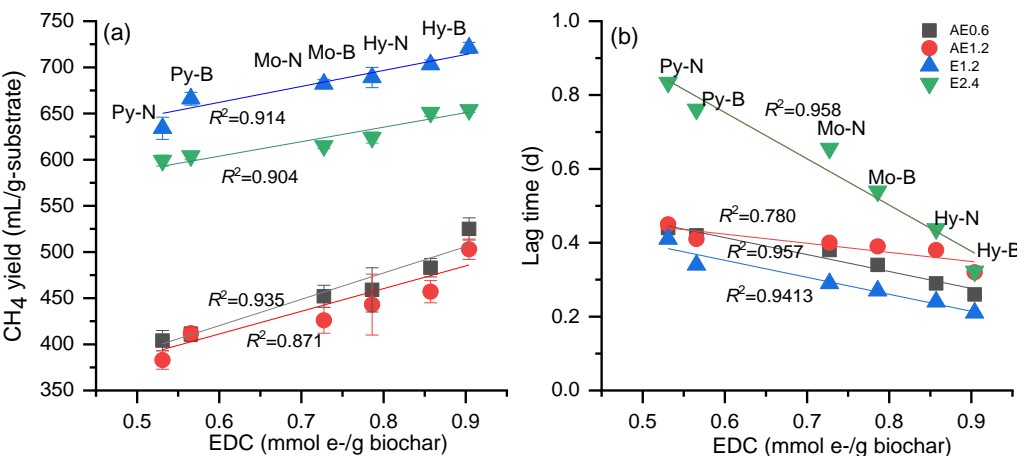

**Figure 6.** (**a**) Correlation between biochar EDC and methane yield and (**b**) correlation between biochar EDC and lag period.

### 3.4. Microbial Community Analysis

The methanogenic performance depends on the symbiotic and intertrophic activities of acetogenic bacteria and methanogenic bacteria. The change in the microbial community can provide clues for improving the methane production performance of the anaerobic digestion system. A variety of bacteria are involved in the microbial community, and it has been pointed out that a variety of microorganisms in the genus *Geobacillus* (*G. Metallireducens*, *G. PickeringII*, and *G. Lovleyi*) have the ability to use ethanol as an electron donor for growth and metabolism [25]. *Geobacter* is a kind of iron-respiration bacteria that is widely distributed in anaerobic environments. It can efficiently obtain electrons from organic compounds [26]. *Geobacter* spp. has been reported to play an important role in DIET, either directly through extracellular pili or using additional conductive materials [27]. In 2010, researchers firstly identified the involvement of *Geobacter* spp. in DIET in a symbiotic environment, which showed that *Geobacter metallireducens* could deliver electrons from ethanol to *Geobacterium sulfureducens* through conductive pili [28,29]. In addition, some researchers have pointed out the important role of *Geobacter* spp. in anaerobic digestion studies. For example, Rotaru et al. showed that *Geobacter* spp. can transfer electrons to methanogens [29,30].

In order to explore whether the addition of biochar will have an impact on the microbial community propogation, the suspended sludge and the biofilm tightly adhered to the biochar were sampled for the microbial community analysis. The distribution of the archaeal and bacterial communities is shown in Figure 7. There was a certain difference in the distribution of the bacterial microbial community between the biofilm and suspended sludge, but the difference in the archaeal community was minor. In all the samples,

*Methanolinea* and *Methanobacterium* were predominant (32.6–40.0%), while *Methanosaeta*, *Methanoesarcina*, and *Methanospirillum* were 2.9–8.7%. The relative abundance of *Geobacter* in biofilm was higher than that of suspended sludge. These results indicated that the addition of biochar could affect the bacterial community distribution tightly conjugated to biochar, but had little effect on the microbial community of the suspended sludge.

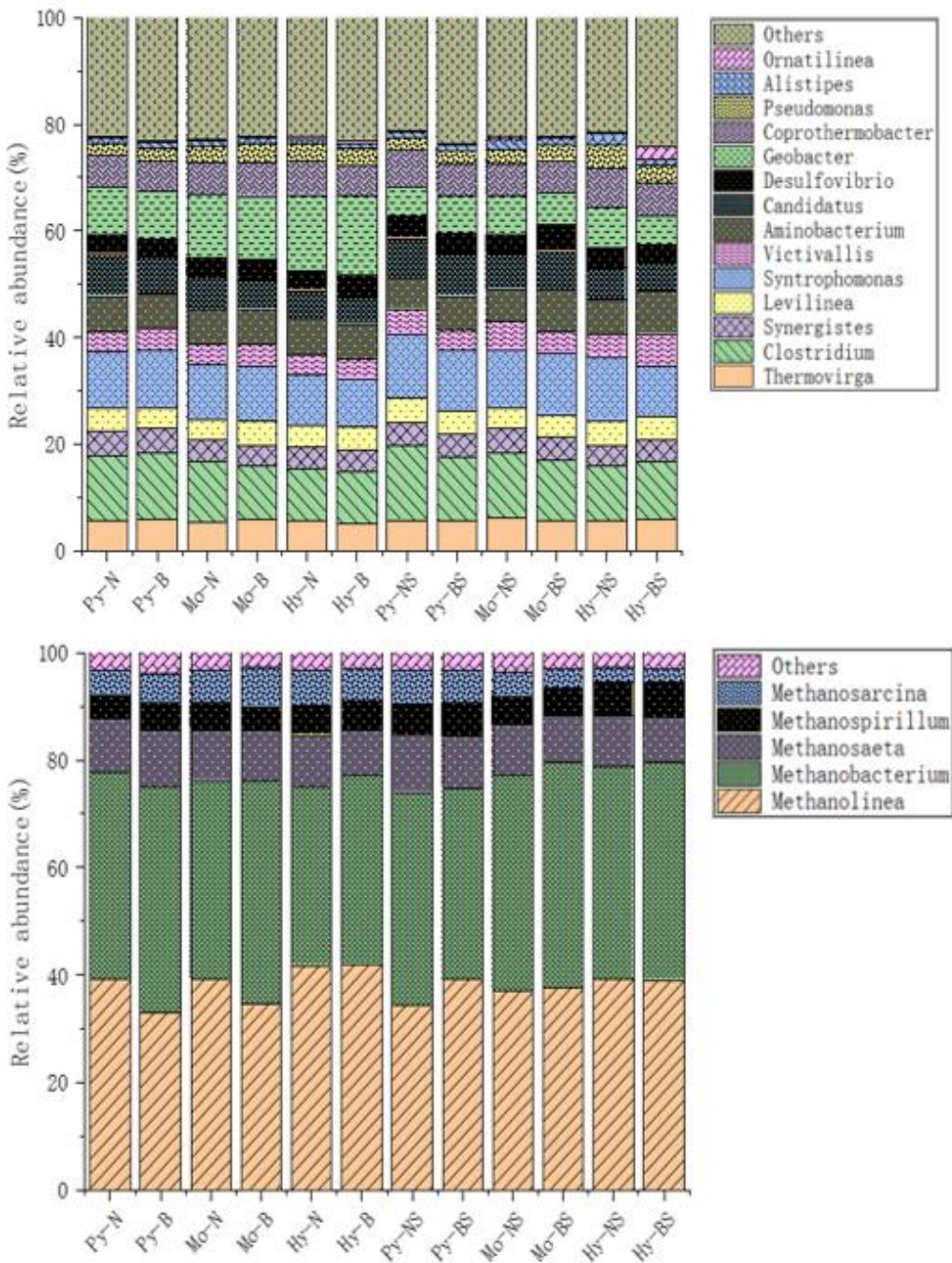

**Figure 7.** Distribution of attached and suspended microbial communities on the surface of biochar.

The increase in the relative abundance of *Geobacillus* sp. in this experiment indicates that anaerobic digestion in this experiment was accompanied by DIET, which accelerated the methanogenesis of the system and enhanced the efficiency of anaerobic digestion for methane production. In addition, the increase in the relative abundance of *Geobacillus* was related to the type of biochar, the order of which is hydrochar > modified pyrochar > pyrochar. Furthermore, the highest abundance of *Geobacillus* sp. was found in the experimental group with bio-aged hydrochar (i.e., Hy-B). It can be speculated that the EDC

of biochar may play a role in the enrichment of *Geobacillus* sp. In addition, it was found that the bio-aging process had little effect on the selective enrichment of *Geobacillus* sp. Thus, bio-aged biochar groups had a better methanogenic performance than that of the new biochar groups, which is presumed to be caused by a greater reliance on stronger electron exchange capabilities and the stronger microbial affinity of the bio-aging biochar.

## 4. Conclusions

This study investigated the promoted anaerobic conversion of acetic acid and ethanol substrates to biomethane by biochar. Based on the anaerobic digestion experiment, the gas production performance of the biochar group was better than that of the control group, and the gas production performance of hydrochar was better than that of pyrochar. The bio-aging process was proven to improve the methane yield from the same substrate. By quantifying the electron exchange capacity of various kinds of biochar, the methane yield was positively correlated to their EDC value. Furthermore, the methane generation performance of the bio-aged biochar was generally higher than that of the new biochar, which may be due to the stronger microbial affinity and increased EDC during the bio-aging process. Thus, this study confirms the sustainability of biochar in promoting anaerobic digestion and the potential for biochar reuse in anaerobic reactors.

**Author Contributions:** Conceptualization, S.X. and H.L.; methodology, Y.J.; validation, Y.R., Y.J. and J.Z.; investigation, Y.R.; data curation, M.L.; writing—original draft preparation, Y.R.; writing—review and editing, S.X. and X.Z.; visualization, Y.J.; supervision, H.L.; project administration, S.X. All authors have read and agreed to the published version of the manuscript.

**Funding:** This research received no external funding.

**Institutional Review Board Statement:** Not applicable.

**Informed Consent Statement:** Not applicable.

**Data Availability Statement:** The data presented in this study are available on request from the corresponding author.

**Conflicts of Interest:** The authors declare no conflict of interest.

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
