# Peer review of "Straw Biochar-Facilitated Methanogenesis from Acetic Acid and Ethanol: Correlation with Electron Exchange Capacity"

_fermentation, doi:10.3390/fermentation9070584_

Round 1

Reviewer 1 Report

1. A recommended title: "Straw biochar facilitated methanogenesis from acetic acid and ethanol: Correlation with electron exchange capacity"

2. Line 11. >> "Straw biochar prepared with 3 methods (..."

3. Line 21. >> "... O-containing"

4. Is there any other effect causing the differences among experiment groups?

5. Line 133.  >> " ... V4 region" as 515F & 806R 

6. Eq.2~4. Reaction equations can be typed in normal text to avoid getting italic.

7. Fig.1. AE groups were quite fast in the first day. More points in the initial days would be preferable.

8. Fig.3. All points started with their respective designed concentrations at Day 0. The detected concentrations would be preferable.

9. Fig.6. Those correlations can be further described with the slopes after  linear fitting.

Minor editing of English language is recommended.

Author Response

Response to reviewer’s comment

Thank the reviewer for the positive and valuable comments on this paper. According to the nice suggestions, the authors have made extensive corrections to this manuscript and the detailed response and revisions are listed below. The changes are highlighted with red color in the revised manuscript.

  1. A recommended title: "Straw biochar facilitated methanogenesis from acetic acid and ethanol:Correlation with electron exchange capacity".

Answer: Thank you for your suggestions. The title has been changed according to the suggestion. (Page 1).

  1. Line 11. >> "Straw biochar prepared with 3 methods (..."

Answer: Thank you for your suggestions. Line 11 has been revised according to the suggestion, “Straw biochar prepared with three methods (i.e., pyrochar, HNO3 modified pyrochar and hydrochar) were added to the anaerobic digestion system”.

Line 21. >> "... o-containing"

Answer: Thank you for your suggestions. The cacography has been corrected, “oxygen- containing functional groups” (Page 1, Line 21).

  1. Is there any other effect causing the differences among experiment groups?

Answer: As reported in the literature, various physicochemical properties of biochar will affect the performance of anaerobic digestion, such as the biochar alkalinity, specific surface area and electric conductivity. etc. Based on the

However, the impact of the electron exchange capacity (ECC) of biochar on anaerobic digestion has been rarely studied. Thus, this study focused on the differentiated effects on methanogenesis from acetic acid and ethanol by straw biochar with different EEC value. Based on the results of our previous studies (Xu et al., 2020), the higher BET was found in pyrochar and modified pyrochar as compared to the hydrochar, nevertheless the correlation of methane yield and the biochar with a higher specific surface area and adsorption capacity was weak. Furthermore, the methanogenic reaction kinetics were evaluated in this manuscript with batch study, in which the substrate was almost consumed after 7 days operation. In this case, the influence of specific surface area on methanogenesis could be ignored. As for the influence of other properties, we would like to continue our exploration in future.

Xu S.Y., Wang C., Duan Y., Wong J.* Impact of pyrochar and hydrochar derived from digestate on the co-digestion of sewage sludge and swine manure, Bioresource Technology, 2020, 314, 123730.

  1. Line 133. >> " ... V4 region" as 515F & 806R

Answer: Thank you for your suggestions. It is corrected in Line 133.

  1. Eq.2~4. Reaction equations can be typed in normal text to avoid getting italic.

Answer: Thank you for your suggestion. The reaction equations have been modified. Page 4, Line 149-151).

  1. Fig.1. AE groups were quite fast in the first day. More points in the initial days would be preferable.

Answer: The authors have also noticed this phenomenon. This result indicated that the inoculum had good adaptability to substrates during the early laboratory domestication. Unfortunately, due to insufficient consideration in the experimental design, these data within the first day was not sampled 1. In the subsequent experiment, we will pay attention to this issue and make improvements. (Page 5, Line 306).

  1. Fig.3. All points started with their respective designed concentrations at Day 0. The detected concentrations would be preferable.

Answer:  Thank you for your suggestions. We have checked the raw data, difference between the detected concentration and the set concentration was small, which has also been updated in the figures.

  1. Fig.6. Those correlations can be further described with the slopes after linear fitting.

Answer: Thank you for your suggestions. Fig. 6 was revised with the linear fitting. Related description and discussion are added in the main text.

Comments on the Quality of English Language

  1. Minor editing of English language is recommended.

Answer: We have double-checked the English to polish the language throughout the manuscript.

Reviewer 2 Report

Authors reported an interesting issue of biochar-mediation of anaerobic digestion of acetic acid and ethanol. The paper should be considered for publication after extensive editing of English language. In the form presented now the paper is extremely difficult to read, e.g.

+ in the headspace of hrdrochar reactor

+ surface o-containing functional groups

+ In the worldwide, the massive

+ In the context of China's rapid economic development and transformation, agricultural production and rural lifestyle have undergone new changes, and traditional problems such as regional, seasonal and structural excess of straw still exist, leading to severe challenges in the utilization of agricultural straw resources in China [5].

+ However, an aerobic digestion system also has many disadvantages, such as slow reaction rate, odor generation, neutral environment and low concentration of wastewater is not applicable.

+ researchers have concerned over

+ the specific functional groups of biochar was increased and leading to a higher electron donating capacity (EDC) [11].

+ both the microbial symbiosisand physichmical properties of biochars

+ ...

Authors reported an interesting issue of biochar-mediation of anaerobic digestion of acetic acid and ethanol. The paper should be considered for publication after extensive editing of English language. In the form presented now the paper is extremely difficult to read.

Author Response

Response to reviewer’s comment

Overall comment: Authors reported an interesting issue of biochar-mediation of anaerobic digestion of acetic acid and ethanol. The paper should be considered for publication after extensive editing of English language. In the form presented now the paper is extremely difficult to read.

Answer: The positive evaluation from the reviewer is greatly apprecitated. We have double-checked the English to polish the language throughout the manuscript.

  1. in the headspace of hrdrochar reactor

Answer: The spelling error has been corrected. “in the headspace of hydrochar reactor” (Line 19).

  1. + surface o-containing functional groups

Answer: The spelling error has been revised, “surface oxygen-containing functional groups” (Line 21).

  1. + In the worldwide, the massive

Answer: This sentence has been revised. “In worldwide, the extensive use” (Line 30).

  1. + In the context of China's rapid economic development and transformation, agricultural production and rural lifestyle have undergone new changes, and traditional problems such as regional, seasonal and structural excess of straw still exist, leading to severe challenges in the utilization of agricultural straw resources in China [5].

Answer: This sentence has been revised. “In recent years, although the utilization rate of agricultural straws is increasing in China due to the governmental guidelines and related industry development, a large amount of straw is directly returned to the field or utilized as fertilizer. With the gradual improvement of the requirements of carbon neutral management in agriculture, the need to develop high-value utilization has become increasingly prominent.” (Line 30-33).

  1. + However, an aerobic digestion system also has many disadvantages, such as slow reaction rate, odor generation, neutral environment and low concentration of wastewater is not applicable.

Answer: This sentence has been revised. “However, an anaerobic digestion system also has many disadvantages, such as slow reaction rate and low methane yield[5].” (Line 47-49).

  1. + researchers have concerned over

Answer: This sentence has been revised. “researchers are concerned about”. (Line 50-51).

  1. + the specific functional groups of biochar was increased and leading to a higher electron donating capacity (EDC) [11].

Answer: This sentence has been revised to “ the phenolic and lactonic groups on biochar were increased, leading to a higher value of electron donating capacity (EDC)” (Line 63).

  1. + both the microbial symbiosis and physichmical properties of biochars

Answer: This sentence has been revised. “both the microbial symbiosis and physicochemical properties of biochars” (Page 2, Line 73-74).

Round 2

Reviewer 2 Report

I accept the revision